# The Small Auxin-Up RNA *SAUR10* Is Involved in the Promotion of Seedling Growth in Rice

**DOI:** 10.3390/plants12223880

**Published:** 2023-11-17

**Authors:** Xiaolong Huang, Zhanhua Lu, Lisheng Zhai, Na Li, Huiqing Yan

**Affiliations:** 1School of Life Sciences, Guizhou Normal University, Guiyang 550001, China; huangxiaolong@gznu.edu.cn (X.H.); zhallisheng94@163.com (L.Z.); 232100100397@gznu.edu.cn (N.L.); 2Key Laboratory of Plant Physiology and Development Regulation, Guizhou Normal University, Guiyang 550001, China; 3Laboratory of State Forestry Administration on Biodiversity Conservation in Mountainous Karst Area of Southwestern China, Guizhou Normal University, Guiyang 550001, China; 4Rice Research Institute, Guangdong Academy of Agricultural Sciences, Guangzhou 510640, China; aaslzh@163.com

**Keywords:** SAUR, *Oryza sativa*, auxin, *YUCCA*, *PIN*

## Abstract

Small auxin-up-regulated RNAs (SAURs) are genes rapidly activated in response to auxin hormones, significantly affecting plant growth and development. However, there is limited information available about the specific functions of SAURs in rice due to the presence of extensive redundant genes. In this study, we found that *OsSAUR10* contains a conserved downstream element in its 3′ untranslated region that causes its transcripts to be unstable, ultimately leading to the immediate degradation of the mRNA in rice. In our investigation, we discovered that OsSAUR10 is located in the plasma membrane, and its expression is regulated in a tissue-specific, developmental, and hormone-dependent manner. Additionally, we created *ossaur10* mutants using the CRISPR/Cas9 method, which resulted in various developmental defects such as dwarfism, narrow internodes, reduced tillers, and lower yield. Moreover, histological observation comparing wild-type and two *ossaur10* mutants revealed that *OsSAUR10* was responsible for cell elongation. However, overexpression of *OsSAUR10* resulted in similar phenotypes to the wild-type. Our research also indicated that *OsSAUR10* plays a role in regulating the expression of two groups of genes involved in auxin biosynthesis (*OsYUCCAs*) and auxin polar transport (*OsPINs*) in rice. Thus, our findings suggest that *OsSAUR10* acts as a positive plant growth regulator by contributing to auxin biosynthesis and polar transport.

## 1. Introduction

Small auxin-up RNA (SAUR) is specific to plants and responsive to auxin application. Several other phytohormones and environmental signals also influence SAUR expression [1]. SAUR plays a crucial role in the dynamic and adaptive growth of plants. It was first discovered in the hypocotyl of soybean, which was found to be auxin-inducible for cell elongation. Since then, SAUR has been extensively studied in *Arabidopsis thaliana* and other plant species [2,3].

The *SAURs* are a sizeable multigene family and are particularly abundant due to remarkable tandem and segmental duplications [4]. Plants contain various *SAUR* members that are arranged in clusters in their genomes. The number of members ranges from 3 (*Anthoceros angustus*) to 308 (*Triticum aestivum*) among the identified species to date [3,5]. For example, Arabidopsis and rice genomes contain 79 and 58 members [6,7]. They have small open reading frames varying from 180 to 540 bp, including a lack of introns, and encode proteins of 9~20 kDa. Each member has the SAUR-specific domain (SSD), which consists of 60 amino acids and confers a high degree of homology, resulting in functional redundancy [8]. However, the lower conservation and variable ranges of N- and C-terminal sequences may be required for the functional divergence among the SAUR family. It is noteworthy that the *SAUR* transcripts and their encoded proteins are unstable and degrade quickly after auxin induction, based on the presence of conserved downstream (DST) destabilizing elements (ATAGAT, GTA, or GAT(N)xGTA) in their 3′ untranslated region (UTR), conferring the instability of the SAUR transcripts [9]. Subcellular localization prediction revealed that the SAUR family members could be localized to the plasma membrane, cytoplasm, chloroplast, nucleus, and chloroplast [7]. For example, both OsSAUR39 and OsSAUR45 are localized to the cytoplasm, whereas AtSAUR36, AtSAUR62, and ZmSAUR2 are in the nucleus [10,11,12,13,14]. AtSAUR32 and AtSAUR53 are localized to both the plasma membrane and nucleus [15,16].

*SAURs* participate in a plethora of developmental processes that have been studied and validated using genetic and molecular tools such as gene editing, RNA interference, or overexpression plants. The first and most prominent function is involved in auxin-mediated cell expansion to regulate plant development by influencing auxin biosynthesis and transport differently. For example, overexpression of *AtSAUR41* and *AtSAUR63* resulted in pleiotropic and favorable auxin-related phenotypes, consisting of longer hypocotyls, enhanced vegetative biomass, promoted lateral root development, expanded petals, and twisted inflorescence stems [17]. Likewise, overexpression of *AtSAUR19* revealed activities in inducing the growth of leaves, stems, and stamen filaments through cell wall expansion and cell elongation, achieved by interacting with the D-clade type 2c phosphatases (PP2C-Ds) protein to stimulate plasma membrane H^+^-ATPases’ activities, thus resulting in increasing osmotic water flow [18,19]. Similarly, *AtSAUR63* was found to have an exclusive expression in root meristems and enhanced expansion by increasing auxin transport [20]. *TaSAUR66-5B* also promoted root growth and increased biomass and grain yields in wheat by enhancing auxin biosynthesis [5]. Defects in the genes mentioned above lead to auxin response and polar transport disruption, thus promoting cell expansion by regulating auxin transport. However, some *SAURs* play antagonistic roles in the process of cell expansion. For example, overexpression of *AtSAUR32* inhibited hypocotyl growth of Arabidopsis [21]. *OsSAUR39* negatively regulated auxin synthesis and transport, since plants with overexpression of *OsSAUR39* exhibited reduced lateral root development and shoot length. Similarly, overexpression of *OsSAUR45* negatively regulated growth and repressed transcripts of the auxin biosynthesis gene flavin-binding monooxygenase (*OsYUCCAs*) and auxin transport gene PIN-FORMED family proteins (*OsPINs*) [10,11,22].

The expression profiles of *SAURs* respond to internal or environmental cues to regulate plant development and growth. First, *SAURs* have distinctive expression patterns in various tissues to fine-tune plant development [23]. For instance, *AtSAUR51* is expressed explicitly in the meristem, expanding leaves, root tips, and lateral root primordia; the maximum expression of *AtSAUR62* is in stamen filaments, petals, sepals, stigmas, styles, pollen tubes, and pollen grains [14]; *AtSAUR63* is significantly expressed in hypocotyls, petioles, cotyledons, and flowers. Moreover, *AtSAUR41* is exclusively expressed in the microenvironment of the root tip as well as the hypocotyl endodermis, whereas *AtSAUR71* and *AtSAUR72* located in the same subbranch of the phylogenetic tree are highly expressed in the hypocotyl and the central column of young roots [17]. In addition, *AtSAUR71* is also expressed in stomata, and its transcript level is altered with the stage of stomatal development [24]. Secondly, most promoters of *SAURs* contain multiple *cis*-acting regulatory elements associated with phytohormones and other signaling pathways, such as ABRE elements, MBS, W-box, and G-box regulated by abscisic acid (ABA), MYB, W-box, bZIP, bHLH, and NAC, respectively [23]. Furthermore, the expressions of *AtSAUR20* and *AtSAUR63* are responsive to auxin signaling in the ARF-BZR-PIF complex. *AtSAUR16*, *AtSAUR50*, and *AtSAUR51* are enhanced in response to zeatin [23]. Finally, the transcript levels of many *SAURs* are circadian, as the DST element is related to circadian control and contributes to the rhythmic expression of SAURs. For example, *AtSAUR63* is highly expressed in the morning but decreases in the afternoon through the upstream control of clock genes [25]. 

Despite plant *SAURs* being functionally characterized, most *SAURs* in rice remain poorly understood. Since *SAURs* are expressed differentially in varied tissues in rice (*Oryza sativa*), it is necessary to explore the functions of *SAURs*. In this study, we conducted knockout of *OsSAUR10* using the clustered regulatory interspaced short palindromic repeats/CRISPR-associated protein 9 (CRISPR/Cas9) editing technique and overexpression of *OsSAUR10*. Our study suggested that *OsSAUR10* might act as a positive regulator of rice growth.

## 2. Results

### 2.1. Structural and Phylogenetic Characterization of OsSAUR10

*OsSAUR10* is 1451 bp in length on the second chromosome, consisting of 501 bp cDNA and 3′ UTR with a conserved DST element. OsSAUR10 contains an auxin-inducible structural domain between 26 amino acids (aa) and 112 aa. The conserved motif of the protein sequence confers its ability to be induced by auxin. The motif has also been found in its orthologs (Appendix A). We constructed a phylogenetic tree using the protein sequences from triticale, maize, rice, celery, and Arabidopsis to investigate the phylogenetic relationship of SAUR10. Based on the analysis, the phylogenetic tree was divided into four clusters: group I, II, III, and IV. Among the four clades, group IV contains multiple SAUR members (Appendix A). OsSAUR10 falls into group II, and a more detailed phylogenetic analysis with other SAUR members demonstrated that OsSAUR10 shared a close homolog with ZmSAUR54 (Figure 1A). Furthermore, the most homologous protein in Arabidopsis to OsSAUR10 is AtSAUR53, which has been reported to positively regulate apical hook development and tissue elongation in Arabidopsis [16].

To investigate the detailed subcellular location of the OsSAUR10, a *GFP* vector fused with *OsSAUR10* was expressed transiently in rice leaf protoplasts. The results showed that the protoplasts transformed with *35S::GFP*, as a positive control, displayed green fluorescent signals in the entire cell, including the membrane, cytoplasm, and nucleus under an exciting 488 nm light (Figure 1B). By contrast, when transformed with *35S::OsSAUR10:GFP*, the GFP signals were only distributed in the cell membrane, precisely at the stained position using chloromethyl-benzamidodialkyl carbocyanine (CM-Dil), which was exclusively stained in the plasma membrane. Hence, we inferred that OsSAUR10 was localized in the plasma membrane.

### 2.2. OsSAUR10 Had a Distinctive Expression Pattern and Was Expressed in a Circadian Manner

To explore the expression pattern of *OsSAUR10*, various organs of *Oryza sativa* subsp. Japonica cv. Zhonghua11 (ZH11) at different periods were collected for qRT-PCR analysis, including seeds, seedlings 7 days after germination (DAGs), roots, leaves, sheaths, panicles at stage 3 and stage 8, and florets (Figure 2A). *OsSAUR10* was higher in young seedlings, especially roots, in which levels were 14.24 times and 15.86 times higher than those expressed in the matured seeds, individually. Moreover, *OsSAUR10* was highly expressed in the early developing stage of the panicle, and its transcript degraded immediately after further development, based on the fact that the transcript in Panicle 3 was 11.44 times than that in Panicle 8, suggesting that *OsSAUR10* might play a crucial role in seedling development in the early stages.

Meanwhile, the promoter activity of *OsSAUR10* was analyzed using the β-Glucuronidase (GUS) reporter. A total of 2000 bp promoter sequences of *OsSAUR10* were amplified using primers listed in Appendix A for the GUS assay. The qRT-PCR results illustrated that strong staining signals of 7 DAGs seedlings were observed in the whole root, suggesting the accumulation of *OsSAUR10* (Figure 2B), especially in root hair and elongation regions. We further magnified the root and observed staining signals in the apical meristem compared with the root cap, suggesting its predominant expression in elongating tissues of roots (Figure 2C,D). The histological observation further suggested that *OsSAUR10* was strongly expressed in meristematic tissues, including parenchymal cells, primordia, and endosperms with the well-differentiated zones, indicating that *OsSAUR10* might promote cell elongation (Figure 2E,F). However, some weak signals were noticed in leaves (Figure 2G) and root hairs (Figure 2H).

The expression profiles of *OsSAUR10* were further investigated during the early germinal periods. Young seedlings were collected at intervals of 12 h from 0 to 120 h after seed germination (HAGs) and used to test the transcripts of *OsSAUR10* (Figure 2I). *OsSAUR10* transcript was significantly decreased from 0 to 24 HAGs according to the qRT-PCR analysis. It rapidly elevated and reached its peak at 60 HAGs. Interestingly, the expression was immediately reduced at 96 HAGs and subsequently enhanced again until 120 HAGs, suggesting that *OsSAUR10* was expressed in a circadian manner.

### 2.3. OsSAUR10 Positively Responded to Varied Phytohormone Signals

The promoter of *OsSAUR10* was found to reveal the potential regulated factors. Apart from the conserved sequences, the *cis*-acting elements of *OsSAUR10* have been identified to be associated with transcriptional factors (bZIP, MYB, and AP2) (Appendix A). The results also revealed that *OsSAUR10* could be responsive to auxin, ABA, GA, and cytokinin (CK) signals.

We further investigated the expression of *OsSAUR10* in 7 DAGs seedlings under various treatments, including auxin, polar auxin inhibitors (PAT), ABA, SA, and 6-BA. The results demonstrated that *OsSAUR10* was remarkably enhanced by auxin, its analogs, or the PAT inhibitors. Among them, indole-3-acetic acid (IAA) was the conspicuous factor in influencing *OsSAUR10*, followed by naphthaleneacetic acid (NAA) and 2,4-dichlorophenoxyacetic acid (2,4-D). Besides auxin influx inhibitors naphthoxyacetic acid (NOA) and 3-chloro-4-hydroxyphenylacetic acid (CHPAA), the auxin efflux inhibitor 2,3,5-triiodobenzoic acid (TIBA) enhanced the transcripts of *OsSAUR10* despite no significant change with auxin efflux inhibitor naphthylthalamic acid (NPA). In addition, *OsSAUR10* was also significantly increased after ABA and gibberellin acid (GA_3_) treatments and did not differ with salicylic acid (SA) and 6-benzyladenine (6-BA) (Figure 3A). We further used a histochemical staining assay, and GUS staining could be found in the cell elongation zone of the root. After the application of IAA and ABA for five hours, more obvious staining signals were found in the same part, and no significant change was observed when the root was treated with 6-BA, indicating that IAA and ABA substantially enhanced *OsSAUR10,* according to the qRT-PCR results (Figure 3B). 

### 2.4. The OsSAUR10 Mutants Impeded Seedling Growth

To evaluate the role of *OsSAUR10* in rice development, a CRISPR-Cas9 construct expressing gRNA targeted to the exon of *OsSAUR10* was generated and transformed into rice. Two knockout lines were obtained and proved to belong to homozygous mutant plants based on the sequencing analysis of DNA fragments using *OsSAUR10* genomic-specific primers. The *ossaur10#6* was produced by a guanine deletion at 192 within the exon, and the other *ossaur10#15* was generated by deleting five bases from 192 to 196 (Figure 4A). 

The seedlings of *ossaur10* mutants displayed obvious morphological defects, such as reduced whole-plant height, decreased shoot length, narrower leaves, and smaller panicles. The whole seedlings of the two knockout lines were significantly dwarfed (Figure 4B). A comparison of four internode lengths of the stems (first, second, third, and fourth) of *ossaur10* was conducted. They exhibited prominently shorter and slower growth than ZH11, especially the first internode (Figure 4C). We conducted histological observation using periodic acid–Schiff (PAS) reagent staining and found that the primordia of leaf sheaths (triangle) and shoot apical stems (asterisk) were substantially larger in wild-type compared to *ossaur10* mutants, suggesting that *OsSAUR10* might promote cell size and improve the growth and development of rice (Figure 4D). We gathered the 7 DAGs seedlings and analyzed their agronomic characteristics. Our results revealed that two *ossaur10* mutants had a lower seedling height, 12.89% and 15.74% less than the wild-type (Figure 4E). Additionally, the root lengths of two *ossaur10* mutants decreased by 17.26% and 8.59%, respectively (Figure 4F). The overall plant height and the number of tillers were also reduced by 3.59% and 26.67%, respectively (Figure 4G,J). Despite no significant change in matured seeds and seed setting rate between wild-type and two *ossaur10* mutants (Appendix A and Figure 4H), the findings revealed that the average seed germination rates of two *ossaur10* mutants were only 83% and 70%, respectively, which were both lower compared to the 85% germination rate of the wild-type (Figure 4I). Therefore, it can be deduced that *OsSAUR10* plays a critical positive role in seedling growth and is responsible for cell elongation.

### 2.5. OsSAUR10-Overexpression Displayed No Apparent Effect on Seedling Growth

We further transformed *35S::OsSAUR10* into ZH11 and obtained *OsSAUR10*-overexpressing plants. Two overexpression lines, *OsSAUR10-ox#3* and *OsSAUR10-ox#*12, were compared with the wild-type and showed no significant phenotype divergences (Figure 5A). The qRT-PCR results confirmed that the *OsSAUR10* transcripts in *OsSAUR10-ox#*3 and *OsSAUR10-ox#*12 were 12.84 times and 7.54 times increased in the wild-type, respectively (Figure 5B). However, we found no substantial changes between the overexpression and wild-type lines when measuring the seedling height and root length of 7 DAGs seedlings using at least fifteen plants (Figure 5C,D). 

### 2.6. OsSAUR10 Influenced Auxin Synthesis and Transport

Since SAURs act as early response genes of auxin, they can potentially affect auxin biosynthesis and polar transport. Herein, we explored the effects of *OsSAUR10* on the transcripts of the auxin synthesis *OsYUCCA* gene and auxin efflux carrier *OsPIN* gene family. The qRT-PCR showed that only expression of *OsYUCCA1* among the seven family members was significantly promoted in the *OsSAUR10-ox* lines. However, the expressions of the other four members, *OsYUCCA3*, *OsYUCCA4*, *OsYUCCA6*, and *OsYUCCA7*, were remarkedly promoted in *ossaur10* lines, despite no noticeable change in the remaining two *OsYUCCAs* (Figure 6A). Regarding auxin transport, the expressions of 12 *OsPIN* members were determined. We obtained that the transcripts of *OsPIN1a*, *OsPIN1b*, *OsPIN5a*, *OsPIN5b*, *OsPIN5c*, and *OsPIN10b* were significantly enhanced in the knockout line, whereas only *OsPIN5b* and *OsPIN8* were increased in *OsSAUR10-ox* lines (Figure 6B). The results demonstrated that the up-regulation of most OsPIN members could influence the knockout of *ossaur10* lines. 

## 3. Discussion

### 3.1. OsSAUR10 Positively Regulates Seedling Growth in Rice

OsSAUR10 shares a common conserved core region responsive to auxin, thus positively regulating seedling growth [3]. Similarly, *AtSAUR41* and *AtSAUR71*, in the same clade as *OsSAUR10*, promote cell growth and organ elongation in Arabidopsis [24]. Despite overexpression of *OsSAUR39* and *OsSAUR45* being reported to inhibit plant growth, here, we revealed that knockout of other SAURs resulted in the same phenotype by affecting auxin synthesis and carriers through the gene expression of *OsYUCCAs* and *OsPINs* [10,11]. Both knockout lines exhibited defective phenotypes, such as dwarf seedlings, decreased tiller number, lower germination rate, and lower yield compared with wild-type, indicating that *OsSAUR10*, different from the two previously reported *SAUR* members, was a positive regulator of rice growth and development.

The phylogenetic relationships of SAUR proteins from five species showed that four groups were clustered. OsSAUR10, situated in cluster II, displayed distant evolution from OsSAUR45 and OsSAUR39, categorized into group III. Likewise, an early study showed that OsSAUR10 was distantly separated from OsSAUR45 and OsSAUR39 among 58 members of rice [6]. Additionally, from a perspective of genomic distribution, OsSAUR10 localized on the second chromosome, whereas OsSAUR45 and OsSAUR39 were located on the ninth chromosome. Moreover, subcellular localization results showed that OsSAUR10 localized in the plasma membrane, whereas OsSAUR39 and OsSAUR45 were both in the cytoplasm, indicating that they may take part in different biological processes to be responsive to auxin signals. 

### 3.2. Tissue-Specific and Circadian-Regulated OsSAUR10 Is Inducible by Varied Phytohormones 

*OsSAUR10* exhibited diverse expression profiles throughout plant development in varied tissues. An early study showed it was highly expressed in 3 DAGs seedlings [3]. *OsSAUR10* was also significantly expressed in the 7 DAGs seedlings in our research, especially in the region of root elongation and panicles in the early development stage. Furthermore, the histological observation displayed strong GUS staining signals in parenchymal cells, the endodermis, and the epidermis (Figure 2), thus suggesting that tissue-specific expression of *OsSAUR10* enhanced cell expansion and division to promote rice growth [23]. 

The 3′UTR of *OsSAUR10* contains a highly conserved DST element that determines the stability of its transcripts. The putative DST element regulated the *OsSAUR10* transcript circadian in the early germination stage. Moreover, two *OsSAUR10-ox* lines exhibited similar agronomic traits to the wild-type, possibly due to the DST element, which confer the unstable *OsSAUR10* and degradation within minutes after transcription. Previous research has indicated that the overexpression of AtSAURs containing the DST element leads to phenotypes that are less severe when compared to those where the DST element is absent [13,23]. 

*OsSAUR10* is involved in different hormone response pathways. It has been found that gibberellin treatment degrades DELLA, leading to the release of BRASSINAZOLE-RESISTANT1 (BZR1) and ARF6, which facilitates the triggering of downstream SAURs [26,27]. Our results also showed that *OsSAUR10* was remarkably increased by auxin and GA3 application, and the promotion might be through the ARF-BZR-PIF signaling module [28]. In addition, its transcript was also substantially enhanced by ABA, suggesting crosstalk between auxin and ABA. Similarly, *AtSAUR41* and *AtSAUR32* were also ABA-inducible, strengthening the plant’s resilience to harsh environmental conditions by harnessing the potential of cell expansion modulation, including drought and salt tolerance [15,29]. The decreased seed germination and dwarf seedlings in the *ossaur10* mutants may result from the altered hormones, which impede seedling development. Several reports in the past revealed the interplay between auxin and the other phytohormones, and their pathways impinged on each other. In rice, the network of inter-regulation between different hormones is very complex. Therefore, the comprehensive interactions result in defective phenotypes in the *ossaur10* mutants. 

### 3.3. OsSAUR10 Regulates Seedlings’ Growth by Impacting Auxin Production and Its Polar Transport

Auxin accumulation and distribution are closely linked to the expression of auxin synthesis genes and their transporter genes [30]. The *YUCCA* gene family, which catalyzes the N-oxidation of tryptamine to form N-hydroxyl tryptamine in vitro, involves embryogenesis and seedling development and is responsible for auxin biosynthesis in shoots [31]. Overexpression of *AtYUCCA2* and *AtYUCCA3* resulted in a similar phenotype with functional redundancy between the different *YUCCA* genes. The functional redundancy was further confirmed through simultaneous analysis of the *AtYUCCA2*, *AtYUCCA4*, and *AtYUCCA6* genes, which were partially overlapping in their expression in the stem tip and flower of Arabidopsis, and the mutation of a single gene did not have a significant phenotype. Among seven *OsYUCCAs* in rice [32], *OsYUCCA1* might be the dominant one that promotes auxin biosynthesis, according to the fact that overexpression of *OsYUCCA1* resulted in a high-growth-hormone phenotype in rice. In contrast, the antisense gene expression increased rice growth hormone insensitivity [33]. Our study found that *OsYUCCA1* was significantly promoted in the overexpression seedlings despite the enhancement of *OsYUCCA3*, *OsYUCCA4*, *OsYUCCA6*, and *OsYUCCA7* in the *ossaur10* lines. This result revealed that *OsSAUR10* promoted auxin biosynthesis mainly through the increased expression of *OsYUCCA1*.

PINs play roles in the polarity of auxin transporter, impacting auxin homeostasis and metabolism, and also belong to a multigene family [34]. Eight members of the PIN family were identified in Arabidopsis. At the same time, different members might play different roles [35]. For example, *PIN5* and *PIN8* have antagonistic/compensatory activity [36]. There are 12 members of the PIN family identified in rice [37]. *OsPIN1a* is highly accumulated in the root base of the stem base and mid-column sheath cells, and suppression of *OsPIN1a* expression confers an increase in the number of tillers [38]. *OsPIN2* takes part in basipetal polar auxin transport, promoting the tillers’ spreading growth [39]. The expression of *OsPIN1a*, *OsPIN1b*, *OsPIN5a*, *OsPIN5b*, *OsPIN5c*, and *OsPIN10b* was significantly up-regulated in the CRISPR lines of *OsSAUR10*, implying the repressive regulation effect of *OsSAUR10* on some *OsPINs*. The increased *OsPINs* might contribute to the decreased number of tillers in *ossaur10* mutants. In summary, we learned that the divergence of seed germination and seedling development is probably altered due to the up-regulation of *OsYUCCA* and varied *OsPIN* expression.

## 4. Materials and Methods

### 4.1. Plant Materials and Growth Conditions

The *ossaur10* seedlings produced through genome editing using the CRISRP-Cas9 and *OsSAUR10-ox* lines were grown in the same place as the wild-type in the following conditions: 60~70% humidity, a photoperiod of 12 h light (28 °C) /12 h dark (24 °C), and 450 μmol/m^2^/s light intensity. For phytohormone treatment, seeds were put on sterile gauze saturated with Yoshida’s nutrient solution and grown in darkness at 37 °C until germination. We collected 7 DAG seedlings and incubated them in a nutrition solution containing 1 μM 2,4-D, 1 μM NAA, 10 μM IAA, 1 μM 6-BA, 10 μM ABA, 1 mM SA, 10 μM GA_3_, and growth hormone inhibitors, including 1 μM Naphthylthalamic acid NPA, 1 μM TIBA, 1 μM NOA, and 1 mM CHPAA. After being treated for five hours, the samples were collected, frozen in liquid nitrogen immediately, and stored at −80 °C. 

### 4.2. Phylogenetic and Promoter Analysis

We used the SAUR protein sequences from Arabidopsis (79), *Oryza sativa* (58), *Amborella trichopoda* (26), *Zea mays* (79), and *Cercis canadensis* (48) listed in Appendix A to construct a phylogenetic tree. Multiple sequence alignment was performed using ClustalW2 (https://www.ebi.ac.uk/Tools/msa/clustalw2/, accessed on 16 November 2023). The phylogenic tree was displayed using the maximum likelihood method in MEGA 11 (https://www.megasoftware.net, accessed on 16 November 2023) and bootstraps with 1000 replicates to evaluate the reliability of nodes. Then, we used the iTOL (https://itol.embl.de/, accessed on 16 November 2023) online tool to annotate and visualize the resulting tree. 

The 2 kb genomic DNA sequences upstream of the transcriptional initiation site of *OsSAUR10* were retrieved from the Rice Genome Annotation Project database (http://rice.uga.edu/, accessed on 16 November 2023). Afterward, the promoter sequences were submitted into the New PLACE Web database (https://www.dna.affrc.go.jp/PLACE/?action=newplace, accessed on 16 November 2023) to obtain *cis*-acting regulatory elements.

### 4.3. Subcellular Localization Assay

To analyze subcellular localization, a GFP reporter tag was constructed. The open reading frame of *OsSAUR10* was amplified and fused into the N-terminal start of GFP under the control of the CaMV 35S promoter (*35S::OsSAUR10:*GFP) of pM999 [40]. The reconstruction vector was then introduced via polyethylene glycol (PEG)-mediated transient transformation into rice protoplasts, and fusion protein localization was monitored in the following 24 h [41]. Meanwhile, the plasma membrane was labeled with CM-Dil at a concentration of 10 μmol/L for 15~20 min. Fluorescence images were captured using the LSM710 laser scanning microscope (Carl Zeiss, Jena, Germany) with a 20 × objective, 500 gain, 488 nm excitation wavelength, and 507 nm emission wavelength. 

### 4.4. GUS Staining

The 2 kb promoter sequence of *OsSAUR10* was amplified from the rice genomic DNA using primers (Appendix A), digested using *Bam*H I and *Pst* I (TaKaRa, Kusatsu, Japan), and inserted into the pDX2181 binary vector containing the GUS reporter gene [42]. This vector was transformed into ZH11 plants via an Agrobacterium-mediated transformation procedure [43]. Different tissues of seedlings at 7 DAGs were collected and put into a 2 mL tube, and reaction solution containing 20 mL 100 mM sodium phosphate, pH 7.0, 0.5 mM potassium ferrocyanide, 1 mL 0.5 M EDTA, 0.5 mM potassium ferricyanide, 5 mg chloramphenicol, 50 mg X-Gluc, 0.1% Triton X-100, 10 mL methanol, and ddH_2_O up to 50 mL was added. After being treated for 24 h at 37 °C, X-Gluc stained tissues were decolorized in solution (glacial acetic acid: ethanol: H_2_O = 1:3:6) for 2 h, washed 2~3 times, and examined using a stereo microscope (SZX16, Olympus, Hamburg, Japan).

### 4.5. RNA Isolation and qRT-PCR Analysis

Total RNA extracted from various tissues and seedlings of wild-type and *OsSAUR10-ox* was reverse-transcribed into cDNA using an RT-PCR Kit^®^ with an oligo dT-adaptor primer (TaKaRa, Kusatsu, Japan). The qRT-PCR was performed in a LightCycler480 instrument (Roche, Rotkreuz, Switzerland) with the FastStart DNA Master SYBR Green I kit. The procedure was performed at 95 °C for 5 min, 40 cycles of 95 °C for 25 s, annealing at 57 °C for 15 s, 72 °C for 25 s, and 4 °C for 10 min. To estimate the expression levels relative to the control, we calculated ΔΔCt and then analyzed the data using the 2^−ΔΔCt^ method. We used a ubiquitin gene (*LOC_Os03g13170*) as an internal standard gene to normalize cDNA starting amounts. All samples were subjected to three biological repeats.

### 4.6. Vector Construction and Transgenic Rice Plants

A nucleotide sequence with specific gRNA was designed from the exon sequence of *OsSAUR10* (*LOC_Os02g30810*) and transformed into a pCXUN-CAS9 backbone vector [44]. A 501 bp cDNA fragment of *OsSAUR10* was amplified and inserted into the pU2301-Flag using gateway (Thermo Fisher, Waltham, MA, USA) [45]. These constructed vectors were transformed into *Agrobacterium tumefaciens* EHA105 and subsequently transferred into ZH11 calli as previously described [46]. Different transgenic lines were obtained, and their agronomic traits were investigated.

### 4.7. Histological Observation

The roots were cut into small pieces, fixed in FAA (10% formalin (*v*/*v*), 5% acetic acid, 50% ethanol) for vacuum pumping, and stained in hematoxylin for at least 48 h before use. After rinsing the samples with distilled water twice, they were dehydrated using a series of ethanol solutions (30%, 50%, 70%, 85%, and 95% for 1 h each, before being placed in 100% ethanol for 2 h). Afterward, the samples were made transparent using a series of chloroform solutions (1/5, 2/5, 3/5, 4/5, and 5/5 for 1 h each) and then impregnated with broken paraffin wax for 3 days. Finally, the samples were embedded in melted paraffin wax, and sections of samples (8 mm) were made using an RM2265 microtome (Leica, Wetzlar, Germany). In addition, shoot apex samples of wild-type and *ossaur10* mutants were made for histological observation and stained with PAS reagent. After washing with distilled water three times, all slides were examined using a BX53 microscope (Olympus, Hamburg, Germany).

### 4.8. Statistical Analysis

The data were represented as mean ± standard deviation and compared using analysis of variance (ANOVA) followed by a comparison with Tukey’s test using the Statistical Program for Social Sciences (SPSS) program version 20.0 (SPSS Inc., Chicago, IL, USA). * denotes significant difference at *p* < 0.05 and (**) represents significant difference at *p* < 0.01. 

## Figures and Tables

**Figure 1 plants-12-03880-f001:**
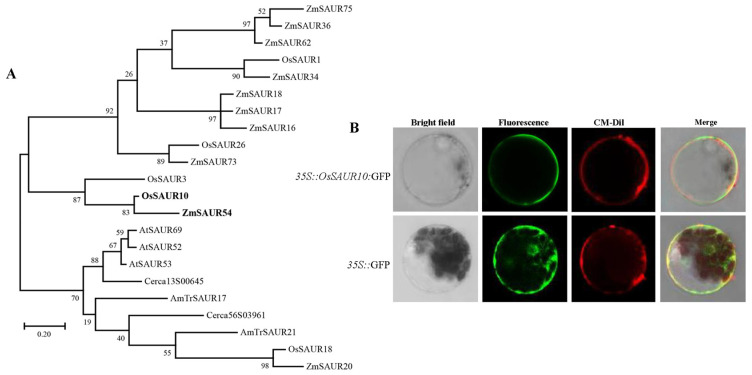
(**A**) Phylogenetic tree of OsSAUR10. Numbers next to each node represent confidence percentages. (**B**) Subcellular location of OsSAUR10. *35S::OsSAUR10:GFP* and the control *35S::GFP* were transiently expressed in rice protoplasts individually. Bar = 10 μm.

**Figure 2 plants-12-03880-f002:**
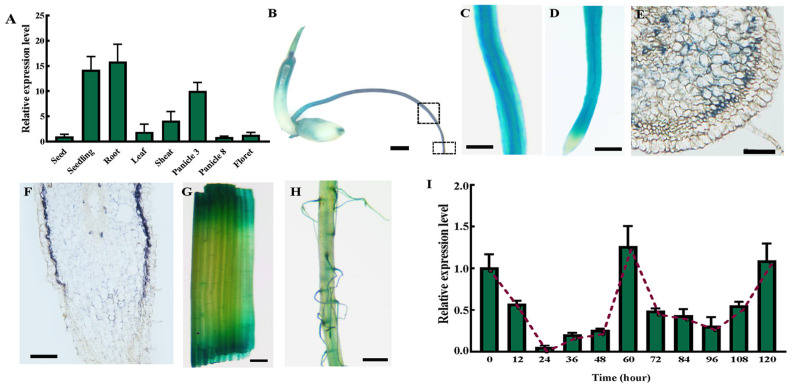
The expression profiles of *OsSAUR10*. (**A**) qRT-PCR analysis of *OsSAUR10* in different tissues of rice. (**B**) GUS staining of seedling 7 days after germination, and the squares individually indicate the elongation regions and the root cap. (**C**) The enlargement of elongation regions and (**D**) the root cap of the root. (**E**) The cross-section of histological observation, (**F**) the longitudinal section of histological observation, (**G**) leaf, and (**H**) root hair. (**I**) *OsSAUR10* transcripts at intervals of 12 h from 0 to 120 HAGs (hours after germination). (**B**–**D**,**G**,**H**) bar = 0.1 cm; (**E**,**F**) bar = 100 μm.

**Figure 3 plants-12-03880-f003:**
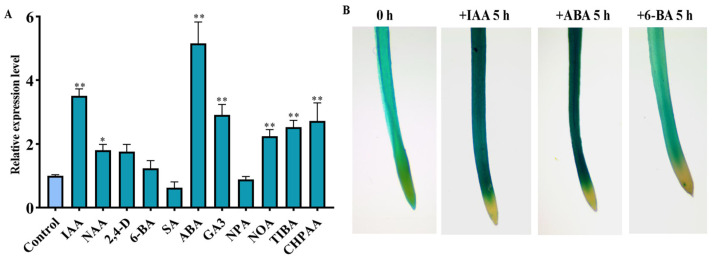
The inducible responses of *OsSAUR10* to various phytohormone treatments. (**A**) Analysis of *OsSAUR10* expressions in 7 DAGs seedlings treated with different phytohormones. * *p* < 0.05, ** *p* < 0.01. (**B**) GUS staining of 7 DAGs seedlings treated with 10 μM IAA, 10 μM ABA, and 1 μM 6-BA for 5 h.

**Figure 4 plants-12-03880-f004:**
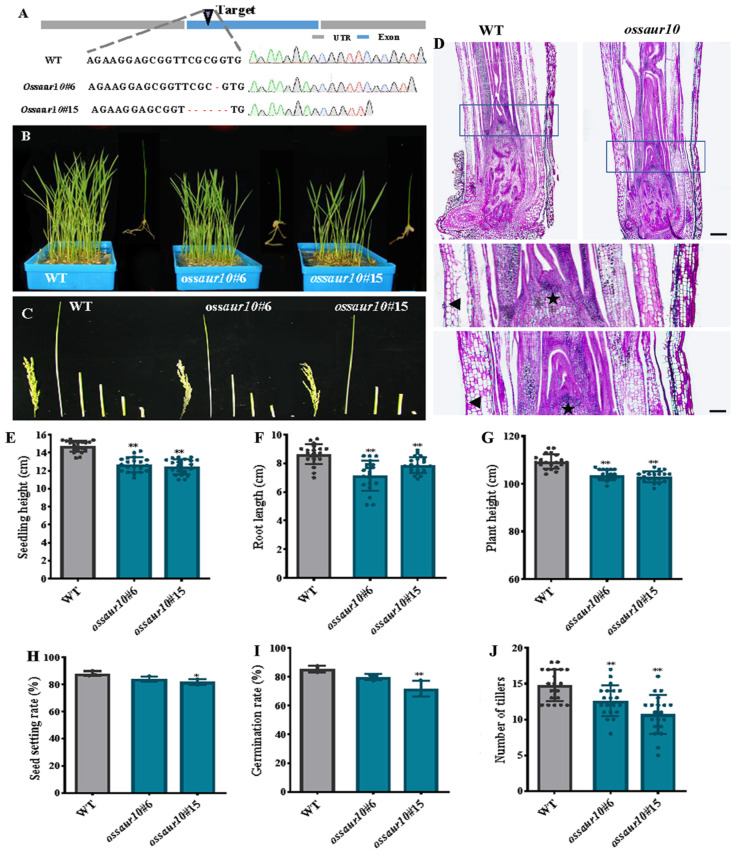
Phenotypic analysis of *ossaur10* mutants. (**A**) The *ossaur10* mutants were generated using CRISPR-Cas9. The upper panel shows the gene model of *OsSAUR10*, indicating the positions of single nucleotide deletions in the *ossaur10* mutant lines. The lower panel displays the mutation sites of *ossaur10* compared to the wild-type (WT) sequences. Four color indicate four nucleotides (**B**) Seedlings of 60 DAGs wild-type and *ossaur10* plants. (**C**) Panicles and different internode lengths of wild-type and *ossaur10* plants. (**D**) Longitudinal sections of histological observation through shoot apex from wild-type and *ossaur10* mutants using periodic acid–Schiff staining (PAS) reagent and enlargement of the blue box. The asterisk denotes the shoot apical meristem, and the triangle represents leaf sheath cells, bar = 100 μm. Wild-type and *ossaur10* plants after heading. (**E**) Seedling height of wild-type and *ossaur10* plants, as well as (**F**) root length, (**G**) plant height, (**H**) seed setting rate, (**I**) germination rate, and (**J**) the number of tillers. (*) denotes significant difference at *p* < 0.05, (**) represents significant difference at *p* < 0.01 and the circle dot indicates the specific agronomic value of each seedling.

**Figure 5 plants-12-03880-f005:**
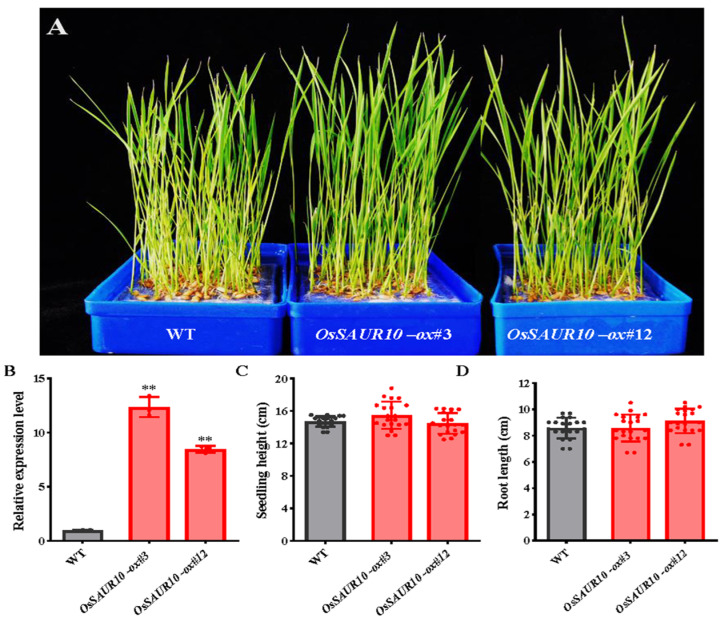
*OsSAUR10*-ox seedlings shared the same growth as wild-type (WT). (**A**) The phenotype of 60 DAGs seedlings. (**B**) The relative expression level, (**C**) seedling height, and (**D**) root length of wild-type and *OsSAUR10-ox* seedlings. (**) represents significant difference at *p* < 0.01 and the circle dot indicates the specific agronomic value of each seedling.

**Figure 6 plants-12-03880-f006:**
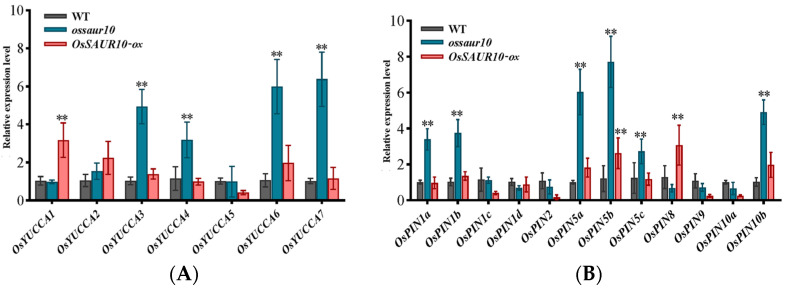
qRT-PCR analysis of auxin-related genes in three genotypes. (**A**) Transcripts of auxin growth hormone synthesis *OsYUCCA* gene. (**B**) Auxin efflux carrier *OsPINs* gene family expression in WT, *ossaur10*, and *OsSAUR10-ox* seedlings. (**) represents significant difference at *p* < 0.01.

## Data Availability

Data are contained within the article and Appendix A.

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
