# Peer review of "The Small Auxin-Up RNA SAUR10 Is Involved in the Promotion of Seedling Growth in Rice"

_plants, 2023, doi:10.3390/plants12223880_

Round 1
Reviewer 1 Report
Comments and Suggestions for Authors
The authors have conducted a functional study of a SAUR family gene OsSAUR10 and validated their potentially positive role in the promotion of seedling growth using transgenic, GUS, histochemical, sub-cellular localization, hormone treatments and qPCR methods.
The sub-cellular localization experiment used by authors to determine the plasma membrane localization is not convincing. For instance, for 35S::GFP in Fig 1B, fluorescence is distributed in the plasma membrane, which contradicts the description in lines 127-131. Moreover, CM-Dil for GFP looks like an edited image.
The
authors state that the SAUR family is highly redundant in rice. Have you
checked the potential off-targets in the homologs in gene editing? Also,
it is worth exploring the changes in cell size in mutants that might potentially
be caused due to auxin imbalance.
Figure
2. Please indicate the developmental stage of seeds used to evaluate the
expression of OsSAUR10.
Line
216-218: The reduction in seed rate in the mutant (Fig 4E) does not have
statistical support. Also, the observed phenotypic difference in ossaur10
mutants are not derived from ‘seedlings’. What is the effect of mutation on
mature seed size?
Line
218: Typo “seed stings”.
Line
223-230: Authors can clearly indicate respective reduction in the agronomic
traits in the main text. In the present description, it is hard to follow what
is the percentage reduction for each KO line. Please rewrite the section.
Line
406: What is the fragment size of the upstream sequence used for GUS assay?
Comments on the Quality of English Language
A professional English edit would significantly improve the manuscript's integrity.
Reviewer 2 Report
Comments and Suggestions for Authors
this research about A small auxin-up RNA SAUR10 is involved in the promotion of 2 seedling growth in rice. they analyzed OsSAUR10 consisting of the conserved downstream element in its 3’ untranslated region, destabilizing the transcripts of OsSAUR10 and contributing to its mRNA immediate degradation in rice. OsSAUR10 was localized in the plasma membrane. The tissue-specific and developmentally regulated expression of SAUR10 18 was also inducible by varied phytohormones. they found OsSAUR10 regulated expressions of 22 auxin biosynthesis flavin-binding monooxygenase family proteins (OsYUCCAs) and auxin efflux carrier PIN-FORMED family (OsPIN) in rice. Our results suggested that OsSAUR10, involved in auxin biosynthesis and transports, acted as a positive plant growth regulator.
the idea is clear and good and the experiments for studying this idea is ok
but i still have some comments
1- abstract need added more about their results
introduction need to rewrite again and make your aim more clear.
your results
your figs is not clear
plz check your statistic again like Fig 3 A)
Discussion
plz added paragraph in your introduction and Discussion about (SAUR) why its important to plants
Comments on the Quality of English Language
its ok
Reviewer 3 Report
Comments and Suggestions for Authors
The manuscript "A small auxin-up RNA SAUR10 is involved in the promotion of seedling growth in rice" is well-written and presents a thorough investigation into the role of the OsSAUR10 protein in rice seedling growth. The authors have employed a comprehensive array of methods, showcasing depth in their research approach. The paper is recommendable for acceptance following some minor revisions, as detailed below.
Line 18: Change "SAUR10" to "OsSAUR10" to maintain uniformity and clarity
Line 155: Based on my understanding, the authors evaluated promoter activity. It might be clearer to state, "The promoter activity of OsSAUR10 was analyzed using the β-Glucuronidase (GUS) reporter."
Lines 210 and 214: It's mentioned that there's a "first exon," but the gene is intronless. Therefore, references to a "first exon" are incorrect.
Line 249: Could the authors clarify how they distinguished between the expression levels of native and transgenic OsSAUR10?
Figure 2: The label on the Y-axis and the size of the scale markers should be consistent and unified.
Supplementary Tables: The numbering is inconsistent in the text; Table S3 is referenced before Tables S1 and S2.
Supplementary Figures: Consider including captions to provide context and explanation.
Plasmids: The origin of all plasmids used in the study, namely pM999, pDX2181, and pU2301-Flag, should be detailed.
Promoter Region Length: Could the authors specify the length of the promoter region that was fused with the GUS reporter? Is this length consistent with what was used for the in silico analysis of cis-acting regulatory elements (2 kb)?
In summary, this manuscript offers valuable insights into the role of OsSAUR10 in rice seedling growth. The minor revisions suggested are intended to enhance clarity and provide a broader perspective to readers. Once these revisions are addressed, I recommend the paper for acceptance in the journal.
Round 2
Reviewer 1 Report
Comments and Suggestions for Authors
The authors have addressed all of my concerns and suggestions.
Reviewer 2 Report
Comments and Suggestions for Authors
the authors did all suggestion that i asked